# A Hyperledger Fabric-Based System Framework for Healthcare Data Management

**Qianyu Wang and Shaowen Qin ***

College of Science and Engineering, Flinders University, Tonsley, SA 5042, Australia;
Qianyu.wang@flinders.edu.au
* Correspondence: shaowen.qin@flinders.edu.au

**Abstract:** This study examined the requirements for privacy-preserving and interoperability in healthcare data sharing and proposed a blockchain-based solution. The Hyperledger Fabric framework was adopted due to its enterprise-grade data processing capabilities and enhanced privacy protection functions. In addition to the Fabric's built-in privacy-preserving functions, healthcare data-specific smart contracts with hierarchical access control were developed to strengthen privacy protection in data sharing. The proposed healthcare data-sharing framework is based on Australian medical practices with the aim to upgrade, rather than to replace, the existing data management models. The outcome of this study demonstrates the feasibility of applying blockchain technology to improve privacy-preservation while enhancing interoperability in healthcare data management.

**Keywords:** blockchain; Hyperledger Fabric; healthcare data management; medical data

## 1. Introduction

The digitization of medical records nowadays has led to increased concerns regarding privacy and security of healthcare information. In Australia, the government and medical institutions have gradually established E-health services and national electronic health records over the past two decades. In 2012, the national electronic medical records platform Personally Controlled Electronic Health Record (PCEHR) was launched. It was renamed *My Health Record* in 2016 [1]. As of October 2021, more than 91% of residents in Australia have a *My Health Record*. More than 99% of pharmacies, 99% of general practitioners, and 97% of public hospitals have registered on this digital medical platform. 4.93 billion documents have been uploaded by healthcare providers and consumers [2]. Australia has become a leader in the fields of medical information transmission and electronic medical record creation. Although the healthcare information platform has been established, nearly 10% of Australians have opted out due to privacy concerns [2]. Indeed, in terms of privacy-preservation, many features of this platform still need to be improved, such as preventing data loss, unauthorized access, and leakage of personal privacy. In addition, it is crucial for patient's peace of mind if transparent data accessing and sharing records are easily available.

In terms of data processing, the storage, verification, synchronization, and sharing of medical records have always been challenges that are difficult to address [3]. When healthcare providers and researchers need to access and share healthcare data, they are under strict policy and technological constraints, which means a substantial amount of time and resources must be spent on conducting permission review and data verification. In most cases, the databases of each hospital are independently managed; each platform has its own set of standards and there is a lack of motivation and incentive for sharing data between different healthcare organizations [4]. These problems have hindered health information sharing which subsequently affects the provision of more cost-effective healthcare services and data-driven solutions on a large scale.

With the recent rapid development of blockchain technology, a large amount of effort has been invested in exploring its applications in various industries. In healthcare, for example, it is proposed that records such as diagnosis, prescriptions, and payments can be stored on the blockchain, as well as all informed parties. The technology can also be used to trace medical supply and drug information for better safety control [5]. An important application of blockchain technology in the health information field is to allow the preservation and sharing of medical information while enabling interoperability [6]. It has been proposed that the anonymity mechanism of the blockchain can be used to protect patient identity information [7,8]. The introduction of smart contracts and blockchain business frameworks has also improved the feasibility of applications [7].

The main contribution of this paper is that we propose a healthcare data management framework and mechanism that is based in a Hyperledger Fabric blockchain environment. With the support of smart contracts, the improved model has the potential to overcome privacy concerns as well as increase user engagement. Several simulation models were designed for evaluating the feasibility of the proposed framework. Together they demonstrate the scenarios and advantages of using blockchain technology for healthcare data sharing.

The rest of this paper is structured as follows. Section 2 presents a literature review on privacy concerns in the healthcare domain followed by a brief overview of the blockchain technology. Section 3 introduces existing frameworks followed by gap analysis. The system design and implementation of the proposed solution are shown in Section 4. In Section 5, we analyzed the system feasibility and discuss the advantages and limitations of the proposed solution. The conclusions are provided in Section 6.

## 2. Literature Review

### 2.1. Privacy in Healthcare Domains

The definition of privacy is different in different contexts. This paper adopts the views of Price and Cohen (2019), that is, healthcare data privacy violations occur when the wrong actor can access the information; the process of accessing the information violates laws and regulations; or the purpose of the access is inappropriate [9]. The violation of privacy will cause concerns both on personal interests and social ethics.

There have been various attempts to address privacy issues. From the legislative aspect, healthcare privacy in Australia is protected by a combination of Commonwealth, State, and Territory legislation. The Privacy Act 1988 (Privacy Act) regulates the privacy legal obligations that all of the health service providers in the Commonwealth public sector and the national private sector must comply with to protect their patient's health information. Usage, disclosure, modification, and destruction of health records are prescribed in the Australian Privacy Principles (APPs). Most Australian states and territories have equivalent privacy jurisdictions that apply to public and private sector healthcare providers [10].

More legislations have been established to further strengthen the governance of privacy protection of electronic health records. These include the Healthcare Identifiers Act 2010 (HI Act), the My Health Records Act 2012, the My Health Records Rule 2016, and the My Health Records Regulation 2012. These legislations and regulations set out privacy requirements for collecting, using, and disclosing personal information to operate and manage the My Health Record system [11].

From a technology standpoint, researchers developed various privacy-preserving techniques that were aimed at privacy concerns for individuals. Data anonymization techniques, such as data masking [12] and pseudonymization [13], are used to protect private and sensitive information. Data perturbation modifies the original dataset by adding random noise to ensure the statistical properties, but it is generally at the cost of destroying the authenticity and integrity of the original data [14]. Data generalization deliberately removes some of the data to make it less identifiable while retaining a measure of data accuracy. Among noise-based approaches, the most representative method is differential privacy [15] which aims to remove individual characteristics while preserving statistical

characteristics to protect user privacy. To solve problems of privacy disclosure and data publishing, the *k*-anonymity anonymizes certain attribute values in the dataset to ensure that at least $k-1$ pieces of the personal information that are contained in the public data cannot be determined by other information [16]. The *l*-diversity extended the *k*-anonymity model by considering the constraints on sensitive attributes; *t*-closeness ensures the distribution of sensitive information does not exceed the threshold *t* [17]. Another approach is to encode or encrypt the original data through cryptographic methods before calculation, such as garbled circuit [18], homomorphic encryption [19], and secret sharing [20]. The secure multi-party computation (MPC) can obtain data use-value without revealing the original data content [21]. The zero-knowledge proof (ZKP) disallows the verifier from obtaining any additional information other than the result of the judgment [22].

In the privacy-preserving domain, the biggest loophole lies in people's illegal behavior. What needs to be done is to develop more solutions at the technical level, thereby reducing the cost of legal supervision and the cognitive cost of users. In this regard, technical measures must interact with legal requirements as well as economic incentives to improve the situation.

### 2.2. Blockchain

#### 2.2.1. Blockchain 1.0—Bitcoin

Blockchain is the basic technical support of the cryptocurrency system that originated from Bitcoin and has since been developed for many application scenarios with innovative models [7]. This technology offers a decentralized architecture that combines distributed data storage, peer–peer transmission, encryption algorithm, consensus mechanism, distributed ledger, smart contract, and other technologies. As the name suggests, the blockchain's data structure is a chain of blocks that contains data (transactions). Moreover, each block has the hash of its previous block, and the blocks connect through hash pointers. The hash of each block is a unique address that is assigned when the block is created. Due to this structure, any modification in a block will lead to changes of the hashes of all blocks in the blockchain which is unrealistic.

To ensure the security and integrity of the data on the blockchain, a variety of modern cryptographic techniques are used including asymmetric encryption, hash functions, and Merkle tree, etc. [23].

The core advantages of basic blockchain technology are decentralization, transparency, and immutability. It can establish trust in a decentralized system by hash pointer, digital signatures, consensus mechanism, and incentives, achieving peer-to-peer transactions and transmissions. The nodes of the entire system, including humans and machines, could share data freely and securely. Excepting encrypted private information, the data in the blockchain is transparent to all of the nodes (persons) and cannot be modified. Hence, a blockchain provides solutions to the problems of high cost, low efficiency, and insecure data storage and sharing that are common in centralized institutions.

#### 2.2.2. Blockchain 2.0—Smart Contracts

The advantage of blockchain technology has led to more commercial-level application possibilities beyond cryptocurrencies being created. One of the most important developments in the blockchain is the introduction of smart contracts. Smart contracts are similar to traditional paper-based contracts, which are described by computer language and executed by the computer, and the contract can be triggered and executed without the trust of a third party [5]. By utilizing smart contracts that can be executed automatically, the cost of contract signing, execution, and monitoring can be reduced significantly in the network.

Ethereum blockchain first introduced smart contracts to create, confirm, and transfer digital assets and contracts. Its core is the Ethereum virtual machine (EVM), which can execute code of complex algorithms on the blockchain. A complete software protocol is defined by the Ethereum project, which is composed of multiple nodes with the same functions so that there is no distinction between the servers and clients [7]. Through the

design and development of smart contracts, Ethereum can implement various complex programs such as crowdfunding systems, digital assets management, supply chain tracking, and monitoring. As a global open-source platform for the development of decentralized applications, Ethereum has improved the commercial potential of blockchain applications.

With the growth of application demand the Enterprise Ethereum Alliance (EEA), which is a collaborative organization, was established in March 2017 by more than 30 corporate giants. It is committed to collaboratively developing standards and technologies to facilitate the deployment of the Ethereum blockchain with improved privacy, security, and scalability. Its representative projects are Quorum (https://consensys.net/quorum/, accessed on 25 October 2021), and Hyperledger Besu (https://www.hyperledger.org/use/besu, accessed on 25 October 2021).

### 2.2.3. Hyperledger Fabric

Another significant development of commercial blockchain technology is Hyperledger Fabric. The Hyperledger project is an open-source and collaborative project which is sponsored by the Linux Foundation that is aimed at improving cross-industry blockchain technology and creating an enterprise-level distributed ledger architecture and codebase. Fabric is one of the projects in the Hyperledger project. Based on a modular architecture, Hyperledger Fabric supports a series of pluggable components such as consensus and membership services [7]. As an underlying platform for blockchain application development, Hyperledger Fabric supports the implementation of permissioned blockchains with nodes that are committed to reaching consensus or verifying transactions being selected by the central authority. Additionally, Hyperledger Fabric is designed to meet enterprise-level requirements, such as performance, verified identities, and private and confidential transactions.

There are many new concepts and functions in Hyperledger Fabric: (1) Peers that represent the node in the organization, responsible for sorting, maintaining consensus mechanism, endorsing, committing, etc. (2) Chaincode that is equivalent to smart contract. (3) Endorsement policy that allows chaincode to select which peer nodes for participating in the voting part of consensus mechanism. (4) Channel that is the private sub-network in the Fabric to isolate different applications and allow a group of organizations to create a separate transaction ledger. (5) Organization that represents entities such as enterprises and institutions in the Fabric network. (6) The ledger that includes the blockchain and the world state- a series of ordered transactions will be packaged into blockchain, while the world state is continuously updated to record changes of data. (7) Membership Service Provider (MSP) that is responsible for managing certificates and membership, every organization that manages a collaborative enterprise can have its own MSP, there is a global MSP on the channel and each role including peer, orderer, and client maintains a local MSP. With the endorsement policy, the operation of the consensus mechanism is not required to issue tokens and mining. Instead, the consensus is achieved through voting with a specific consensus algorithm that depends on the project. Another new feature of Hyperledger Fabric is that it is private and permissioned. Compared with the open permission system that allows all participants to join the network, the members of the Hyperledger Fabric network ensure privacy by registering with a trusted MSP provider.

Fabric has supported three consensus mechanisms since its release: Solo, Kafka, and Raft. Since Fabric 1.4.1, Hyperledger Fabric officially recommends the Raft consensus mechanism [24]. For the Raft algorithm, each node can only be in one of three states:

- Follower—initially all nodes are in this state. The log of the follower can be rewritten by the leader.
- Candidate—the node in the candidate state will initiate an election. If it receives the approval of the majority members, it will transfer to the leader state. Otherwise, if the candidate finds that the leader has been selected, it will return to the follower state.
- Leader—the node has the right to process client requests and ensure that all followers have the same log copy.

In the Raft algorithm, the concept of term is used to identify the management period of the leader. At the beginning of each term there is an election process, in which one or more candidates are competing to become a leader. The node who wins the election will be the leader for the remaining time in the term and begin to process client requests. The request contains the commands to be executed by the replication state machine. The leader appends the command to its log (this command is still "uncommitted") and broadcast to all of the followers to copy the new log. Until all of the followers finally get a consistent log copy, the leader will execute the command on its own state machine (the command is in the "committed" state) and return the result to the client. After other nodes receive the leader's message, the state machine will run requested commands. Thus, the logs of each node are kept consistent.

## 3. Existing Frameworks

Existing research regarding the applications of blockchain technology in the healthcare field mainly includes information protection, payment, data storage and sharing, data transactions, and drug traceability. The related works are as follows:

In 2015, Zyskind, Nathan, and Pentland combined blockchain and off-chain storage to build a personal data management platform [25]. The privacy concerns were resolved by allowing users to manage access permissions. This study has certain reference significance as healthcare data is also a kind of personal data. However, in some cases, the patient does not have enough medical knowledge to address the permissions, so it is inappropriate to completely manage permissions by patients.

In 2016, MedRec was developed based on Ethereum's smart contracts which connected healthcare providers, allowing comprehensive and credible medical history data to be shared across different institutions [8]. The Healthcare Data Gateway (HDG) combined a traditional database and gateway to manage personal electronic medical data on the blockchain, ensuring security and privacy through access control and secure multi-part computing [26]. As a result, the combination of blockchain and other security technologies can strengthen the protection of patient privacy. It should be noted that the basis for those methods to deal with the ownership of medical data is that the data is completely controlled by the patient. However, this assumption is not in compliance with Australian law which requires medical records be retained for a minimum of seven years by healthcare providers. On the other hand, whether the ownership of medical data only belongs to patients is still worth discussing.

In 2017, more researchers started to look into cloud storage and permissioned blockchain. Dubovitskaya et al. presented a blockchain data sharing framework for the primary care of oncology patients from the medical practice perspective where privacy is protected through symmetric encryption and there are no incentives for malicious behavior [27]. BBDS employed encryption and digital signatures to ensure access control [28]. MeDShare is a data-sharing model between cloud providers using blockchain, where the protection of privacy is demonstrated in access control and the trace of violations [29]. With the introduction of cloud computing, the feasibility of blockchain applications has increased but security issues have become more complex.

In 2018, Ancile utilized smart contracts in Ethereum for access control and interoperability of electronic health records which focused on the ownership rights of patients and proposed the proxy re-encryption method to store private keys remotely [30]. FHIRChain demonstrated a standards-based architecture to ensure secure and scalable clinical data sharing in a blockchain environment, which ensured privacy by keeping sensitive data off-chain and exchanging reference pointers on the chain [31]. Liang et al. proposed a Hyperledger Fabric data sharing scheme in mobile healthcare applications in which data sharing and accessing are determined by the users [32]. In William and Christian's scheme, privacy is achieved from patient-driven interoperability in healthcare [6]. The above research still attempted to achieve privacy-preservation through patient-driven access control policy but are lacking solutions for third-party security issues. Moreover, patient-driven ac-

cess control is difficult to convert from the existing model because it increases the difficulty of data management for doctors, clinics, and hospitals without any obvious rewards.

In 2019, Chen et al. proposed EHRs sharing scheme that was based on the blockchain and suggested that in the healthcare data sharing process, blockchain only acts as a trust mechanism and privacy issues should be completely revolved by a third party [33]. Med-Chain combined blockchain, digest chain, and P2P network to overcome efficiency issues in sharing various types of healthcare data which addressed privacy issues through data generation: disassociate the actual data with the data owner [34]. Healthchain suggested that doctors should also have the right to control their diagnostic data [35]. These studies show that in the current health IT systems, data security and privacy are mainly achieved through multiple encryption technologies. Blockchain technology should not be regarded as "the master key" to change all healthcare data management methods and solve patient privacy concerns.

In 2020, a series of studies considered adopting privacy blockchain, including the Hyperledger Fabric framework, to implement scalable healthcare data management applications. PREHEALTH, a privacy-preserving EHR management solution uses Hyperledger Fabric and Identity Mixer to ensure acceptable scalability and data privacy [36]. Tanwar, Parekh, and Evans proposed a Hyperledger Fabric-based EHR sharing system and its related test environment that was based on Hyperledger composer [37]. The Hyperledger blockchain was suggested as an approach for Intensive Medicine, especially ICU data management, to avoid equipment failure [38]. Another approach to enhance security is the so-called blockchain tree that connects three blockchains together to store different types of data separately [39].

The above studies address privacy issues in healthcare data mainly through using the blockchain to give patients access control and monitor data usage. These models take advantage of the openness, traceability, and tamper proofing of the data on the blockchain, and overcome the shortcomings of traditional doctor-driven data storage and sharing. On the premise of solving the security of data storage and transmission, if the healthcare data access permission can be completely controlled by the patient, it could effectively protect privacy by preventing violated access.

However, there are several shortcomings in the above research:

- Healthcare data is a valuable asset, but its ownership is complex. The law does not specify data ownership and its economic value. For example, a doctors' diagnosis and treatment plan, the flow of people in clinics, the economic profits of private hospitals, etc., can all be reflected in the healthcare data and the value that is generated by these data should not belong to patients only. In other words, the "intelligence of property" problems of healthcare information hinders the patient-driven models.
- The patient-driven data management model does not fit the current situation. The limitation is that most patients do not have the expertise to control what kind of information is shared with whom and when, which could result in reduced efficiency of healthcare services provision. The proposed solutions have not benefited the improvement of healthcare services. In traditional face–face treatments, doctors communicate with patients or refer to medical examinations to understand medical history and condition. Letting the doctor browse a patients' healthcare history data is not necessarily more efficient. On the other hand, for healthcare research institutions, requiring access permissions for each type of data from each patient increases the difficulty of obtaining data. It is also costly for patients to decide on each data access request.
- The current frameworks have not reflected the legal and industry regulations on data retention. In Australia, the minimum timeframe for keeping medical records is seven years for an adult and until 25 for anyone under 18 [10].

## 4. System Design and Implementation

To overcome the above shortcomings, this paper proposes an improved healthcare data sharing scheme targeting the privacy issues of medical practice. In compliance with legal requirements and economic motivation, the proposed scheme uses permissioned blockchain and smart contracts to facilitate information sharing and security controlling.

More specifically, the study has the following aims:

- To provide an improved strategy that introduces blockchain technology and smart contracts into healthcare data management to overcome the challenge of privacy-preservation.
- To develop design and implementation details to demonstrate the feasibility of the proposed strategy.

### 4.1. System Overview

The existing literature has largely used a patient-driven access control policy to overcome privacy issues. A recent innovation by Wu and Chen employed a hierarchical access control approach in the E-medicine system [40]. This approach overcomes the efficiency problem of dynamic access problems and enhances interoperability. Hence, this study decided to adopt this method and apply it to blockchain system design.

After synthesizing the findings, a system design along with the mechanisms of agent interactions during data sharing was proposed. The design is based on the Hyperledger Fabric framework, using its pluggable member management services, consensus, and ordering services to simplify deployment and development. Specifically, the raft consensus mechanism recommended by Hyperledger is adopted. Furthermore, a series of smart contracts are designed to provide privacy-preservation, access control, and other functions.

The proposed scheme is based on the Australian medical process practice (covered by Medicare) and legal requirements. Healthcare data archiving and sharing are necessary parts of the medical practice. In the diagnosis and treatment process, the healthcare data that the practitioner has gathered about a patient could help understand the history and determine the treatment strategies of that patient. Specific examples includes information provided by the patient, x-rays, pathology and other test results, referral letters, prescriptions, and other relevant healthcare information is the main object of this research. Such information has the following characteristics:

- Healthcare data is collected and managed in a timely manner.
- There are multiple types of data that are stored in existing health information systems including text and pictures, as well as handwritten and electronic versions.
- The structure, accuracy, and comprehensiveness of the data depend on the medical service providers.
- The law requires retaining the medical records for at least seven years and it is recommended that all doctors keep the medical data for as long as possible.
- The data is scalable and must be easy to share.
- Healthcare information is important privacy for patients.

Based on the above characteristics, the improved data management scheme is required to (1) follow the traditional medical recording habits and patterns; (2) be integrated with the existing healthcare information systems; (3) improve the convenience of patients and doctors; (4) strengthen the privacy protection strategies, especially for reviewers, process, and purpose.

According to the above requirements, Figure 1 outlines the key system functions for the proposed scheme. In the process of generating, using, and storing healthcare information, there are three stages.

## 1 Generate new healthcare data

The new healthcare data will be generated and recorded by the healthcare service providers during testing, diagnosis, and treatment. The healthcare data will be stored in the local and uploaded to the Fabric network.

Feedback     Generate new data

Local database     Fabric

## 2 Process data

Once data are uploaded to the system, data will be assigned an ID. The ID usage record will be stored in the blockchain, while the original data will be stored in the medical cloud.

ID     Original data

Access key

## 3 Access healthcare data

Many types of user can access healthcare data with corresponding permissions. For privacy and supervision purposes, the history of access to each data will be recorded.

Healthcare data     Usage record

**Figure 1.** System functions in the proposed framework.

### 4.2. User Scenarios

Figure 2 shows a schematic of the Hyperledger Fabric network. In it, clinics and laboratories represent 3 organizations. Each organization has its member service provider (MSP) to manage identities and certificates. Organization 4 provides ordering services for the information in this Fabric network. Doctors, nurses, and laboratory technicians are clients in their organizations; they can upload or use healthcare data in the network by linking with peers. The nodes (peers) in each organization are responsible for maintaining the consensus mechanism, endorsement, and commitment. There are two channels in the diagram to isolate the data. For example, in Figure 2 shows two data separation channels, the test results of a certain patient are shared by clinic 1 and the laboratory in channel 1, and the test results of another patient are shared by clinic 2 and the laboratory.

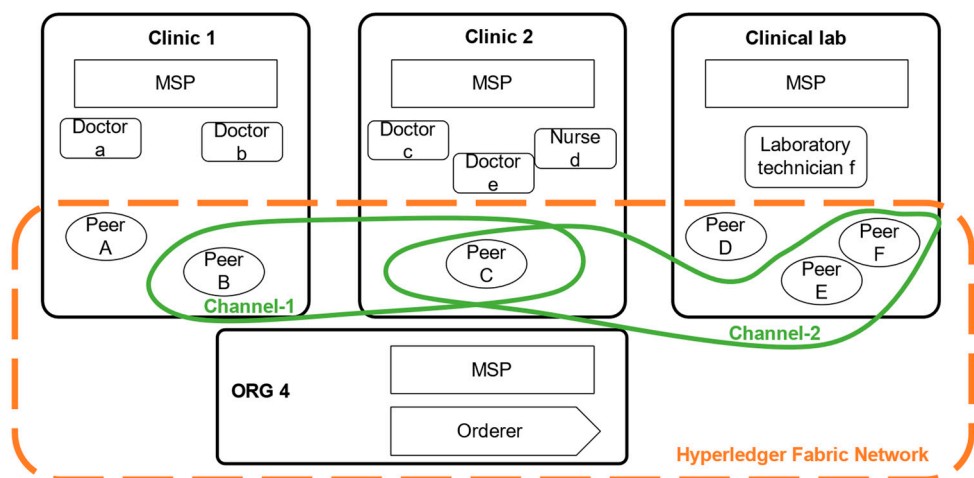

**Figure 2.** Organizations, clients, and peers in Fabric network.

### 4.3. System Entities and Users

Healthcare data. The healthcare information in current medical practices includes the data that is related to a person's medical history, such as electronic health record (EHR), lab results, X-rays, clinical information, prescription history, and notes. In the proposed system, the definition and types of healthcare information are exactly the same as the existing medical practices, which reduces the difficulty of system conversion and upgrade.

In sub-section on implementations, the type of medical data will be limited to test result, diagnosis, and treatment for simplification.

Patient. Traditionally, during diagnosis and treatment, patients usually get immediate feedback on their medical records and need to keep them properly for the subsequent treatment. After treatment, patients have little idea about how their healthcare information is used by the clinic or doctor which has caused many privacy concerns. Under the new scheme, one of the most important functions is to open the access interface for patients themselves, as well as track the usage of data. Since the system would allow patients to completely save all medical records, as well as classify, highlight, and review their healthcare information. On the other hand, user privacy concerns are resolved by allowing patients to track the usage of their healthcare information and identify any illegal access. Each record is attached by the visitor's address, identity, organization information, and access time. Once patients discover illegal access, data usage can be used as evidence to help users protect their rights through legal means.

Medical service provider. Doctors still have the obligation to collect, record, and save medical information, as well as the need to share and forward specific data with other medical service providers. The security of healthcare information in the system not only depends on the doctors' professionalism but also can be supervised by the patients' level of consents. Additionally, the new scheme provides them with a more convenient data management solution by connecting existing different platforms so that data can be shared among different medical institutions without increasing costs. A basic level of consent is that, when the patient authorizes the doctor to provide medical services to them, the doctor obtains permission to reasonably use the patient's healthcare information.

Data processing centre. The virtual logical data processing centre will work as nodes in the blockchain for maintaining the ledger and smart contracts. All applications need to be connected to one of the nodes to achieve the functions of calling data. Its specific work content will be shown in the following implementation part.

Blockchain network. The blockchain network is based on Hyperledger Fabric, which supports a permissioned network where all participants must be authorized. Through the channel, the healthcare data management system achieves the isolation of different services in the blockchain network according to the needs of users, such as clinics and insurance companies. Channels can also be privatized and contain only a specific set of participants. The public key is used to generate encryption certificates that are bound to organizations, institutions, patients, and medical staff. Therefore, setting permission for participants, channels, and access control help to address privacy and confidentiality concerns effectively. Moreover, the storage and sharing of data assets can be achieved through the blockchain; healthcare information is such a data asset, but has not been widely developed. The blockchain also helps to confirm the share of ownership of healthcare data.

Medical cloud. The original healthcare data is fragmented, stored in the cloud, and linked via the blockchain, whereas the usage record of the original data is stored on the blockchain. The two work together to make data access traceable and scalable.

*4.4. System Implementation*

System participant. The establishment and supervision of the entire system are led by the government. Since the public healthcare program is an important part of the common good, the government must provide management, certification, authorization, supervision, and auditing to each system participant during the operation of the system. This paper only discusses basic participant entities including patient, doctor, researcher, and supervisor. Other stakeholders such as commercial insurance companies, pharmacy, and transnational health care industries also have the potential to participate through the Fabric-based consortium blockchain.

Hierarchical access control for privacy issues. The scheme has some special designs to achieve privacy preservation during data sharing. (1) All information usage logs are fed back to and are actively supervised by patients. (2) In the permissioned network,

participation in the system requires authorization. (3) During the consensus process, illegal access will be identified and filtered out. (4) Implementation of the hierarchical access control is through smart contracts, further review the purpose of access, as shown in Figure 3. Similar to Bitcoin which solves the privacy problem by not binding the account address to the physical identity of the address holder [41]. According to the characteristics of healthcare data, someone's medical history, diagnosis, and treatment records are indeed private. However, if the data is not linked to a specific person, this kind of data will not have privacy issues and the cost of obtaining the data legally will also be reduced. The hierarchical access control opens different permissions to different scenarios so that privacy violations can be avoided when the actor, the process, or the purpose of the access is inappropriate.

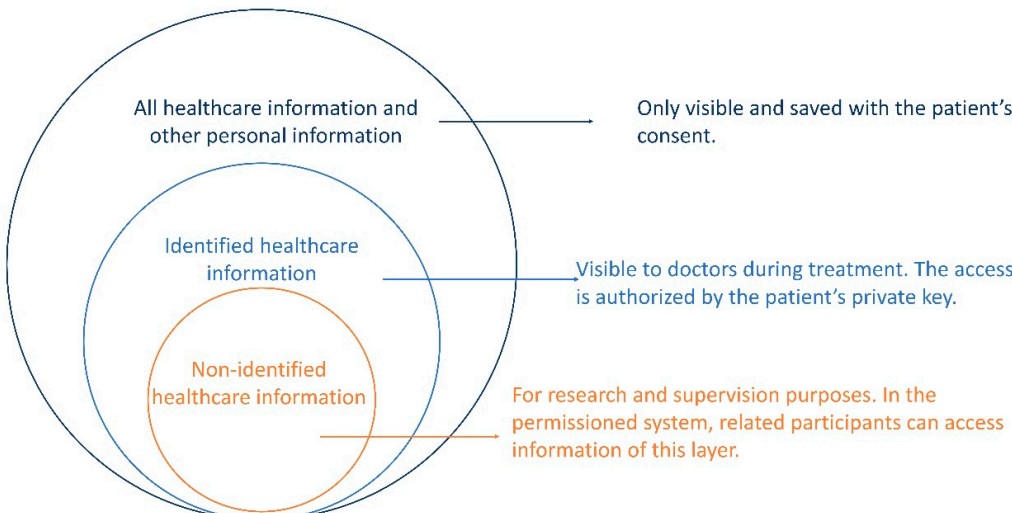

**Figure 3.** Hierarchical access control for privacy-preserving design.

Ledger design. The ledger is composed of the world state and the blockchain. The blockchain is structured as hash-linked blocks. Each block header includes a hash of the current block and a hash value of the prior block so that the data that is stored on the ledger is sequenced and tamper-resistant. The main content that is stored in the block is the index of each original medical record and its usage logs, such as add, query, and read. In this way, the ledger plays the role of a logbook in the system to store the immutable and sequenced healthcare information in blocks. In the implementation of this paper, due to the smaller amount of designed data in the implementation, all data is still stored in the world ledger instead of the medical cloud. The ledger uses the Couch DB database as state database (world state) to achieve the function of searching for cases by keywords. It stores the medical data that we will operate in chaincode in the form of key-value pairs. In the file system of the blockchain, the historical data and the index of the blockchain are stored in Level DB and currently does not support changes. Fabric 2.0x currently supports the Couch DB as database for handling specific business logic. There are three data types that are stored in the ledger. Table 1 shows their details:

Consensus mechanism. In the implementation phase of this article, we use the Raft consensus which is recommended by Fabric officially. In future, other consensus algorithms may become available for deployed through smart contracts.

Incentive mechanism. The Hyperledger Fabric cancels the mining and incentive mechanism because each member in the consortium blockchain must deploy their nodes to ensure data security and user service. The incentive mechanism in this paper refers specifically to the incentive for all stakeholders of the entire scheme. Patients are able to know how and by whom their healthcare information is used, which will increase their participation. Correspondingly, the government can obtain more healthcare data, promote

the development of medical big data, and strengthen the supervision of the healthcare industry. For doctors and medical institutions, the new scheme will not change their medical record archiving practices but it will help understand the patient's complete medical history and improve the convenience of sharing medical records. The department which is most affected by the new scheme is the traditional data centre; they need to be upgraded to achieve node functions. However, this is a reasonable cost of data management, and the government can provide computing centers and medical institutions with incentives such as credit ratings, tax cuts, and job creation. Also, the asset value of healthcare information has not been effectively demonstrated but the parties that generate the data will certainly obtain benefits. The incentives come from the confirmation of the shares of ownership in healthcare information. With the development of intellectual property of medical information, the blockchain has the potential to play significant roles in future healthcare information assets pricing.

**Table 1.** Data types of the ledger.

| Key | Value | Details |
| --- | --- | --- |
| Patient username | Object of patient | Patient's personal information and medical data |
| Medical record number generated by the system. (e.g., case10001, case10002) | Object of case | Each medical record of a patient, simplified to test results, diagnosis, treatment |
| Usage history of medical data (e.g., record10001, record10002) | Usage Record | Medical information users, simplified to patient himself, doctors, regulators. Operations include add new record, read, and append. |

*4.5. Implementation of Smart Contracts*

The implementation uses Go language that was developed under the Ubuntu system. All data are stored in the world ledger. This implementation includes the following functional modules: (1) The patient queries their own personal information and medical history; the patient authorizes the doctor (2) Doctors enter the new medical data into the system, and view patient medical data as well as personal information with authorization. (3) Viewing of medical data for research or supervision purposes can only see data with hidden personal information. (4) All operations on medical data are recorded in the usage record. Together, all of the functional modules will form smart contracts of hierarchical access mechanism.

4.5.1. System Functionalities

1. The patient accesses their own healthcare data: enter the username of the patient, and display the patient's personal information (username, name, address, phone number) and healthcare data (test results, doctor's diagnosis, treatment plan).
2. The patient authorizes the doctor: enter patient username, doctor's username, the patient can click "Ok" to authorize the doctor to access his personal information and medical data for treatment.
3. The doctor enters data into the system: enter the username of the doctor, the username of the patient, test results, the doctor's diagnosis, and the treatment plan, and click "Enter". If there is no permission, it will prompt "Error! Permission denied", if so, it will prompt "Submission complete".
4. The doctor accesses the data: Enter the username of the doctor, the username of the patient, for searching the patient account in the system. If there is no permission, it will prompt "Error! Permission denied". If yes, the patient's personal information (username, name, address, phone number, check and test) and medical information (testing results, doctor's diagnosis, treatment plan) will be displayed.

5. The researcher accesses data: enter the username of the researcher, it will display all medical data that does not contain personal information.
6. The supervisor access data: enter the username of the supervision institution, it will display all medical data that does not contain personal information.
7. Show ID's of the usage records: display all ID usage records.

### 4.5.2. Blockchain Network Deployment

The implementation takes two organizations and four nodes as an example, including orderers and peers (see in Table 2):

**Table 2.** Blockchain network deployment.

| Organization | org1, org2 |
|---|---|
| Peers | peer0, peer1 |
| Anchor nodes | The peer0 node is the anchor node of each organization |
| Channel | emrchanel |
| Chaincode | emrcc |

Since this paper is only a demonstration of the implementation, the deployment of the blockchain adopts the simplest structure. Organizations, nodes, and channels can be increased according to the complexity of smart contracts and business logic. In this demo, the domain name is "medical.com".

### 4.5.3. Data Types

There are three "key-value" data types in the proposed implementation.

The object of patient stores the patient's personal information during initialization. This paper simplifies the types of personal information to address and telephone. Doctors that are authorized by the patient will be recorded in the patient object (see Figure 4).

```
//Patient
Type Patient struct {

    ObjectType          string      `json:"docType"`              // patientObj

    Username            string      `json:"username"`             // username

    Name                string      `json:"name"`                 // full name

    Address             string      `json:"address"`              // address

    Telephone           string      `json:"telephone"`            //telephone number

    Id                  string      `json:"id"`                   //UUid for the patient's medical data

    Cases               []Case      `json:"cases"`                Medical data

    AuthorizedDoctors   []string    `json:"authorized_doctors"`   Username of the authorized

}
```

**Figure 4.** Data type of Patient.

The case is the combination of healthcare data of a patient, simplified to test results, diagnosis, and treatment. The serial number of the case is generated by the system, that is case10001, case10002 (see Figure 5).

The usageRecord records the usage logbook of each case. The serial number of the case is generated by the system, that is record10001, record10002. The usageRecord object is used to trace and supervise healthcare data usage (see Figure 6).

```
//Case
Type Case struct {
    ObjectType      string      `json:"docType"`      // caseObj
    Id              string      `json:"id"`           // UUid for the patient's medical data
    TestResults     string      `json:"test_results"` // test results
    Diagnosis       string      `json:"diagnosis"`    // doctor's diagnosis
    Treatment       string      `json:"treatment"`    //treatment plan
}
```

**Figure 5.** Data type of Case.

```
//UsageRecord
Type UsageRecord struct {
    ObjectType      string      `json:"docType"`      // UsageObj
    Id              string      `json:"id"`           // UUid for the patient's medical data
    Operation       string      `json:"operation"`    // operation type: add, read, append
    Roles           string      `json:"roles"`        // role of operator
    Username        string      `json:"username"`     // username of the operator
    Time            string      `json:"time"`         // time of operation
}
```

**Figure 6.** Data type of usageRecord.

### 4.5.4. Implementation of the Smart Contract

The smart contract is composed of five functions. Figure 7 and Table 3 shows the details of the functions that compose the smart contract of hierarchical access control. While creating new patient accounts in the system, a unique id is generated for each patient by xid (https://github.com/rs/xid, accessed on 25 October 2021), to ensure that the global uniqueness will not be compromised.

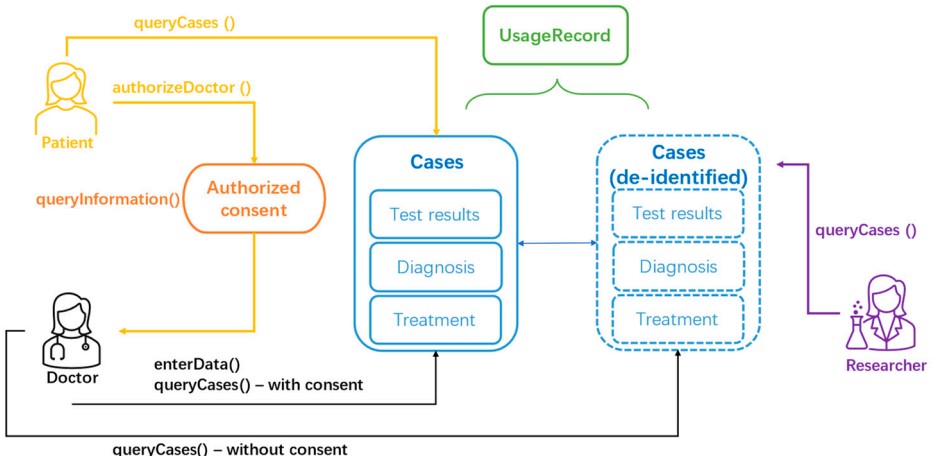

**Figure 7.** System function modules interactions.

Table 3. Functions in the smart contract.

| Functions | Details |
|---|---|
| queryInformation(patient, patientID, time) or queryInformation(doctor, doctorID, patientID, time) | The parameter can be 3 or 4. When the first parameter is "patient", the current time will be automatically obtained in the web (the following time is the same) and passed in together with the patient's ID or username entered on the front-end. When the first parameter is "doctor", the current time will be passed in together with the doctor's ID, patient's ID entered on the front-end. This function first verifies the number of parameters, then determines whether the first parameter is patient or doctor, and goes to different execution contents. The former (patient) obtains the world state value of the patient's username directly from the ledger, records the "read" operation, and finally returns the patient information. The latter (doctor) first obtains the world state value of the patient's username from the ledger, determines whether the doctor's username is in the patient's authorized doctors' array (that is, whether it is authorized), and if so, records the "read" operation and returns patient's personal and medical information. Otherwise, it prompts "unauthorized" |
| authorizeDoctor(patientID, doctorID | The number of parameters is 2. This function verifies the number of parameters, then obtains the world state value of patient's username from the ledger, adds the doctor's username to the patient's authorized doctor array, and then rewrites the modification into the ledger. |
| enterData(doctor, patient, testResults, diagnosis, treatment, currentTime) | This function is used for doctors entering the patient's data into the system. We simplify the entry of each case into the following 6 parameters: the username of doctor, the username of patient, the test results, the doctor's diagnosis, the treatment plan, and current time. This function verifies the number of parameters, and then obtains the world state value of the patient's username from the ledger to determine whether the doctor's username is in the patient's authorized doctor array. If it is, the medical data (test result, diagnosis, treatment) will be written into patient's account by the doctor. Then, the modified patient information and medical case will be rewritten into the ledger. The above "append" operation will also be recorded. If there is no permission, it will prompt "unauthorized". |
| queryCases(role, username, keywords, currentTime) | This function is used to query medical data for research or supervision purposes. It has 4 parameters, which are user category (researcher/regulator), the user's username, query keywords, and the current time. The function first verifies the number of parameters, and then uses rich query to query all cases in the ledger that contain the keyword in the diagnosis category. It returns the content that meets the conditions, and then records the read operation as well as role, username, time. Otherwise, it prompts that there is no information that meets the conditions. |

## 5. Discussion

The blockchain-based healthcare data management scheme that is presented in this paper focuses on privacy preservation and interoperability. This solution is implemented through the privacy-preserving mechanism of the Hyperledger Fabric framework and the smart contract for hierarchical access control. Moreover, due to the advantages of Hyperledger Fabric, such as enterprise-level distributed ledger technology; pluggable functions (e.g., consensus algorithms, components, member management services, etc.); modular; and universal design, this blockchain development platform satisfies a large number of use cases in the healthcare industry. Compared with the mainstream blockchain development platforms, especially enterprise-level frameworks that are based on Ethereum, such as Hyperledger Besu and Quorum, although the cost of deploying and maintaining the Hyperledger Fabric network is higher, companies within large alliances are more inclined to use consortium chains. In the healthcare industry, due to the need to transmit valuable information among hospitals, research institutions, medical groups, large pharmaceutical companies, and insurance companies, the adoption of consortium chains based on the Hyperledger Fabric framework become a more suitable choice. Moreover, due to the high maintenance cost of the Hyperledger Fabric network, it also raises barriers to

illegal acts such as theft of patient privacy information. Based on the above reasons, at present, the proposed solution is more suitable to be implemented through the Hyperledger Fabric framework.

As the core of blockchain technology, the consensus mechanism is an important factor that affects the deployment of the system, the design of business logic, and system performance. Hyperledger Fabric gave up the previously adopted Solo and Kafka consensus because Solo has few applicable scenarios and Kafka needs to deploy Zookeeper externally. At present, Hyperledger Fabric adopts the Raft consensus, which is a Leader-follower model. The leader node is determined by dynamic election and each follower node is the replication of the leader node. The configuration of the Raft consensus is simpler and can better reflect the characteristics of decentralization in the blockchain. It is also suitable for multiple organizations and channels. The proposed solution does not need to deploy a separate consensus mechanism, instead it uses the pluggable Raft consensus provided by Hyperledger Fabric.

Decentralized applications that are based on blockchain technology need to process, store, or update the same ledger at each node, which consumes a large amount of space. The healthcare data storage cycle can be as long as several decades so that the demand for storage space is even higher. The proposed system introduces cloud storage and stores the hash value of the original data in the blockchain, eliminating the need for nodes to have massive storage space. Centralized cloud computing uses traditional and mature data security strategies to ensure that healthcare data can be stored stably and reliably. The blockchain network ensures that data in the system can be traced and cannot be tampered with.

In the patient-driven access control model, all requests for access to healthcare data will be reviewed by the patient and a decision will be made as to whether to authorize access. Although this type of model can protect patient privacy to the utmost extent and prevent unauthorized access requests, the patient does not have enough professional knowledge to determine which data or information should be authorized to improve healthcare service quality; this will cause many negative effects. First, doctors and patients may not be able to reach a consensus on whether to authorize access to certain information, leading to increasing the cost of communication between doctors and patients. Secondly, reviewing and authorizing information one by one will increase data management costs and reduce efficiency and patients may also avoid trouble or be misled to allow all access requests. Moreover, research institutions or government health departments also need to use healthcare data to assist research or policy formulation and patients should not have the right to refuse such reasonable high-privilege access requests. Therefore, the access control model that is based on human decision-making has many shortcomings. It is hoped that access control strategies can be implemented through automated procedures, thereby improving efficiency and reducing costs. The hierarchical access control smart contract that was designed in this paper assigns a security level to each record when information is uploaded to the blockchain network and is linked with the original data through the hash algorithm. Different access requests can be automatically processed, and the system will show them the corresponding security level information according to the audit results. Automated hierarchical access control contracts can reduce the communication costs between patients and information users, avoid false authorizations, and improve the efficiency of access control for all parties.

The feasibility of the proposed healthcare data management solution is embodied not only in the ability to upgrade and reform through existing data sharing schemes but also to benefit the participants. The smart contract for hierarchical access control can automatically grant corresponding access permissions according to the purpose of access to avoid disputes about privacy issues for all participants. For patients, the system can save healthcare records to their client and provide patients with healthcare data usage logs to monitor the purpose of data access. For healthcare service providers, the system has not changed the habit of recording and archiving healthcare services, as well as providing

a more convenient and secure way to access and share data with alternative medical professionals. The usage of healthcare data is saved through the blockchain network, enabling direct supervision of government health departments. Since the proposed solution is based on the Hyperledger Fabric, more data interfaces can be introduced easily to connect more organizations such as research institutions, insurance, pharmaceutical companies, and international medical tourism.

In addition, healthcare data reflects important patterns of national health, living habits, drug use, treatment plans, etc., and is a basis for supporting various participants in the health industry to create value and gain benefits. However, in-kind or monetary based measure of the value of healthcare data has yet to be developed, leading to the original data owner such as patients not being able to obtain tangible benefits at present. The traceability mechanism of the blockchain can lock the ownership of healthcare data, such as medical records, prescriptions, notes, etc., to serve as a basis for benefit distribution when suitable measurement strategies are developed in the future.

The healthcare data management framework that was proposed herein should be considered in light of some limitations. In the proposed data management scheme, the process of patient authorization of doctors can be optimized and automated. To transfer the demo to the actual implementation and evaluation, the data processing and data analysis should be applied to model real and dynamic healthcare datasets. This limitation is apparent in many studies that are based on the application of blockchain technology. Another limitation is that the design and evaluation of the incentive mechanism needs to introduce economics and game theory for further research.

## 6. Conclusions

The non-tamperable and traceable distributed ledger of blockchain technology can provide solutions to the privacy and interoperability issues of the healthcare industry. In particular, the Hyperledger Fabric framework provides deployment and development components for enterprise-grade blockchain applications.

This research proposes a healthcare data management scheme that is based on the Hyperledger Fabric, a hierarchical access control strategy that is realized through Fabric's built-in privacy-preserving mechanism, Raft consensus, and smart contracts. The demo of the deployment and development of the proposed scheme demonstrates the feasibility of the plan. This study is based on Australian medical practice and meets the requirements for the preservation of healthcare data regarding privacy and expiration.

The main contributions of the proposed solution are: (1) Patients are able to supervise access through healthcare data usage logs stored on the blockchain; (2) the smart contract of hierarchical access control can be automatically executed so that requests can only access the data content corresponding to their permissions; and (3) help confirm the ownership of healthcare data and track changes in ownership, and provide a basis for the distribution of original data benefits.

Nonetheless, there are two main limitations in our work. The system demo has been developed for much simplified user scenarios, hence not been able to address many challenging scenarios in a real medical data environment, such as the connection to off-chain databases, as well as analyzing real-time dynamic healthcare datasets. Also, economics and game theory may be included to design and evaluate incentive mechanisms. Immediate future research should focus on the characteristics of healthcare data in the real environment, improving the performance and scalability of the blockchain, including transferring part of tasks that are on the chain to off-chain for processing or improving algorithms.

**Author Contributions:** Conceptualization, Q.W. and S.Q.; writing of original draft, Q.W.; supervision, S.Q.; writing of review and editing, Q.W. and S.Q.; proofreading, S.Q. All authors have read and agreed to the published version of the manuscript.

**Funding:** This research received no external funding.

**Institutional Review Board Statement:** Not applicable.

**Informed Consent Statement:** Not applicable.

**Conflicts of Interest:** The authors declare no conflict of interest.

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
