# Peer review of "A Hyperledger Fabric-Based System Framework for Healthcare Data Management"

_applsci, doi:10.3390/app112411693_

Round 1
Reviewer 1 Report
This paper studies the hyper ledger fabric-based system. The methods are appropriate to the aims of the study. Sufficient information is provided for a capable researcher to reproduce the experiments described. There are no additional experiments that would greatly improve the quality of this paper.
The results are clearly explained and presented in an appropriate format. All the figures and tables are mostly easy to interpret and show essential data that could not be easily summarized in the text.
The published literature is presented as background information and is connected to the specific findings of this study.
Author Response
Dear Editors and Reviewers,
Many thanks for considering my work for Applied Science, and giving us the opportunity to resubmit the paper The Hyperledger Fabric-based System Framework for Healthcare Data Management.
We have made corrections according to the Reviewer’s comments on the Moderate English changes required. Thank you for pointing this out. We have now worked on both language and readability and have also involved native English speakers for language corrections. We really hope that the flow and language level have been substantially improved.
Once again, thank you very much for your comments and suggestions. We hope the revised manuscript is now acceptable to you. If not, we are glad to receive any further feedback which we shall continue to apply our best effort to address.
Sincerely,
Qianyu Wang and Shaowen Qin
28 November 2021
Reviewer 2 Report
This is an interesting contribution to the existing literature, but the paper suffers from several shortcomings listed in the following comments.
- The paper should be checked by a native.
- A discussion section should be added.
- The introduction should be updated by recent researches.
- The novelty and contribution should be clearly bolded.
- The authors should consider some works about Data Analysis that can be applied to model different datasets. For example,
On Comparing and Classifying Several Independent Linear and Non-Linear Regression Models with Symmetric Errors. Symmetry, 11(6), 820.
Comparison of the climate indices based on the relationship between yield loss of rain-fed winter wheat and changes of climate indices using GEE model, Science of The Total Environment 661, 711-722
- It’s better to suggest some subjects for future works.
Best regards,
Author Response
Dear Editors and Reviewers,
Thank you for your decision and constructive comments on our manuscript. We have carefully considered the suggestion of the reviewer and made some changes. Here below we address the questions and suggestions raised by the reviewer.
Point 1: A discussion section should be added
Response 1: We have revised the text format and sub-headings to address your concerns and hope that it is now clearer. Please see pages 15-17 of the revised manuscript, lines 577-666. In the discussion part, we restated the main research results of this article, discussed the reasons for adopting the Hyperledger Fabric framework and Raft consensus, discussed the advantages of the hierarchical access control mechanism, and how this mechanism can improve user participation, and discussed the limitations of this article.
Point 2: The introduction should be updated by recent researches
Response 2: We thank the reviewer for pointing out this issue. We indeed should have included recent data and researches. We have updated the MHR’s statistics, and have added recent researches on Blockchain applications in healthcare. Please see page 1 of the revised manuscript, lines 31-36, and page 2, lines 59-71.
Point 3: The novelty and contribution should be clearly bolded
Response 3: Agree. In the Introduction section and conclusion section, we have accordingly modified the statement about novelty and contribution to emphasize this point. Please see page 2 of the revised manuscript, line 72, and page 17, lines 678-683.
Point 4: The authors should consider some works about Data Analysis that can be applied to model different datasets.
Response 4: It would have been interesting to explore this aspect. However, in the case of our study, it seems slightly out of scope because our project is about healthcare data management and data storage. In the experiments, the hypothetical datasets only assist in demonstrating the design of the system functions. We will consider this point in follow-up research and have involved this point in comment 5.
Point 5:it is better to suggest some subjects for future works.
Response 5: Thank you for pointing this out. We have rewritten this part to the reviewer’s suggestion. Please see the last paragraph in 4. Discussion, page 17, lines 661-666, and the last paragraph in 5. Conclusion, page 18, lines 689.
In addition to the above comments, we have made corrections according to the Reviewer’s comments on the Moderate English changes required. We have now worked on both language and readability and have also involved native English speakers for language corrections. We really hope that the flow and language level have been substantially improved.
Once again, thank you very much for your comments and suggestions. We hope the revised manuscript is now acceptable to you. If not, we are glad to receive any further feedback which we shall continue to apply our best effort to address.
Sincerely,
Qianyu Wang and Shaowen Qin
28 November 2021
Reviewer 3 Report
The paper presents solutions for using the Hyperledger Fabric blockchain for storing and managing patients' health records. The topic of work is important and necessary.
- The statements in lines 290-309 are very controversial, primarily because, often, workers need access to a medical history of several years, as well as a history of prescribed medications and therapies in order to make a correct diagnosis or prescribe the correct medications. Because many drugs lose their effectiveness with prolonged use by the patient.
In addition, the history of illness or disease is personal information that can be used by both insurance companies to increase the cost of the insurer and employers to discriminate on the basis of health or to refuse to hire a patient with certain diseases.
Thus, the use of personal medical data of patients without their explicit consent to this, with a description of all the possible consequences of such use, is unacceptable and unethical.
- Authors analysed several projects based on public blockchain but forgot solutions for personal health data protection based on private blockchain like Blockchain Tree at al.
- In the line 480, the authors mention intellectual property in medical records, but do not specify who will be the owner of this intellectual property - the patient (who can both authorize and prohibit the use of his personal data, and may also require the complete removal of all medical records from the system, as this is provided for by the legislation of many countries); the doctor who entered the data into the system; the clinic where this doctor works or the state that administers the entire system.
- At the beginning of the paragraph 3.5, the functional modules of the proposed system are described, but it is not shown how these modules interact with each other, as well as how the access logs to these records are stored in the blockchain.
Author Response
Dear Editors and Reviewers,
Thank you for your decision and constructive comments on our manuscript. We have carefully considered the suggestion of the reviewer and made some changes. Here below we address the questions and suggestions raised by the reviewer.
Point 1: In lines 290-309, the use of personal medical data of patients without their explicit consent to this, with a description of all the possible consequences of such use, is unacceptable and unethical.
Response 1: You have raised an important point here. However, the solution proposed in this article is based on Australian legislation and tries to assist in protecting patient privacy from a technical perspective. Based on Division 2 16B Permitted health situations in relation to the collection, use, or disclosure of health information of Privacy Act 1988, and Australian Privacy Principle 4 – Dealing with unsolicited personal information, not all collection and use of health data require the patient’s own explicit consent. However, only the professionals who provide medical services and for the purpose of public medical research can view and use health information. It is illegal to provide medical information to the police, Centrelink, ATO, insurance companies, and employers without the permission of the court, and those organizations have no right to view and request it. In Australia, the insurance companies set the contract price based on the cost of historical medical services rather than the history of diseases. The smart contract design in this article is to provide authorized medical service professionals with identity information and provide de-identified information to other medical personnel and research.
Point 2: Authors analyzed several projects based on public blockchain but forget solutions for personal health data protection based on the private blockchain
Response 2: Since the published literature part is presented in chronological order, there was no clear distinction between the public blockchain and the privacy blockchain. We have modified the literature review section to clarify the privacy blockchain-based solutions, including Blockchain Tree, PREHEALTH, Liang et al’s work and Tanwar et al’s work. Thanks to the reviewer’s suggestion that the approach of connecting three blockchains to improve security gave us a lot of inspiration. Please see page 6, lines 302-311.
Point 3: In line 480, do not specify who will be the owner of intellectual property in medical records
Response 3: Currently, the issue of intellectual property in medical records needs to be regulated by law, but the owner of intellectual property must be the shared owner. In the future when medical intellectual properties can create profit, the traceability of blockchain can help clarify the contribution and share ratio. Please see page 17, lines 654-657.
Moreover, in Australian law, the healthcare professionals must keep the patient’s medical records for at least 7 years, and mostly in digital form. Patients cannot require to remove their healthcare records from the national health system.
Point 4: At the beginning of paragraph 3.5, the functional modules of the proposed system are described, but it is not shown how these modules interact with each other, as well as how the access logs to these records are stored in the blockchain.
Response 4: This has been clarified in the revised version of the manuscript, and the figure on system functions interactions has been included in the revised manuscript as new Figure 7.
The access logs are stored in the world ledger. The ledger uses the Couch DB database in the form of key-value pair (usageRecord). Please see page 11, lines 487-492, table 1, and page 14, lines 565-568.
In addition to the above comments, we have made corrections according to the Reviewer’s comments on the Moderate English changes required. We have now worked on both language and readability and have also involved native English speakers for language corrections. We really hope that the flow and language level have been substantially improved.
Once again, thank you very much for your comments and suggestions. We hope the revised manuscript is now acceptable to you. If not, we are glad to receive any further feedback which we shall continue to apply our best effort to address.
Sincerely,
Qianyu Wang and Shaowen Qin
28 November 2021
Round 2
Reviewer 2 Report
The paper can be accepted in current form.
Reviewer 3 Report
The work may be accepted in the present form.